# On Proofs for the Existence of God: Aristotle, Avicenna, and Thomas Aquinas

Xin Liu 

Department of Philosophy, Nanjing University, Nanjing 210023, China; liuxin10.03@nju.edu.cn

**Abstract:** In this paper, I examine Aristotle's cosmological proof of God's existence, Avicenna's metaphysical proof, and Thomas Aquinas's five-way proof. By comparing these proofs, I argue that philosophers and theologians take different approaches to proving God's existence not only because they follow different epistemological principles but, more fundamentally, because they construct different metaphysical frameworks in which God as the Supreme Being plays different roles and is thus clarified differently. The proof of God's existence is also of theological significance. This paper makes an original contribution by showing that, despite Avicenna's harsh criticism, Aquinas returns to Aristotelian cosmological proof. Moreover, Aquinas goes beyond Aristotle by identifying God not only as the First Mover but also as the Creator. The theme of God's existence bridges philosophy and theology, and it also clearly reflects the interplay and mutual influence of Greek philosophy, Arabic Aristotelianism, and Latin Scholastics.

**Keywords:** Aristotle; Avicenna; Thomas Aquinas; God; existence; essence; causality; modality

## 1. Introduction

In this paper, I examine the three proofs of God's existence proposed by Aristotle, Avicenna, and Thomas Aquinas. By reconstructing and analyzing the arguments, I aim to show that proving God's existence is both a philosophical and a theological issue. I argue that philosophers and theologians take different approaches to proving God's existence not only because they follow different epistemological principles but, more fundamentally, because they construct different metaphysical frameworks in which God, the Supreme Being, plays different roles and is thus clarified differently. Moreover, the proof of God's existence is of theological significance. As a theologian, Aquinas abandons Avicenna's metaphysical proof and restores the Aristotelian cosmological proof for theological reasons.

I begin with Aristotle's cosmological proof (Section 2). I argue that Aristotle constructs this cosmological proof, which proceeds from the moved heavenly bodies to God as the First Mover, by following the epistemological principle that proceeds from the sensible to the intelligible. More fundamentally, Aristotle's cosmological proof is constructed within his metaphysical framework of ousiology and aitiology. Specifically, Aristotle's cosmological proof establishes a paradigm for deducing God by argument, inferring cause from effect. Consequently, the later Peripatetics, in discussing related issues, had to target this Aristotelian paradigm, as seen in Avicenna's criticism of it and Aquinas's adherence to it. I then present Avicenna's metaphysical proof (Section 3). I argue that Avicenna constructs this metaphysical proof while criticizing Aristotle's cosmological proof because he not only modifies Aristotle's epistemological principle into the epistemological principle of deduction but, more fundamentally, reconstructs the discipline of metaphysics in a new, syllogistic way. I emphasize that Avicenna constructs his metaphysical proof in the same syllogistic way that he reconstructs metaphysics. Through syllogism, Avicenna establishes a metaphysical proof, which proceeds not from effect to cause, as Aristotle's cosmological proof does, but rather from the universal to the particular. Accordingly, Avicenna deduces God as a special kind of existence, that is, the necessary existence in itself, from universal

existence. Next, I turn to an examination of Aquinas's five-way proof (Section 4). Taking the five-way proof as a whole, I offer a unified interpretation. I argue that, despite Avicenna's harsh criticism, Aquinas returns to the cosmological proof that proceeds from effect to cause not only because he restores the Aristotelian epistemological principle that proceeds from the sensible to the intelligible but also because causality, or more precisely the causal relationship in the creationist sense between the creatures and the Creator, plays a key role in his metaphysical considerations. Just as Aristotle deduces God as the First Mover from the moved heavenly bodies, so Aquinas constructs a five-way proof to deduce God as the Creator from the creatures, whereby the Creator appears either as the First Mover, or the necessary existence in itself, or as the efficient, final, or existential cause of all things. Aquinas also appeals to the cosmological proof for theological reasons. Consistent with Aquinas's cosmological proof, natural theology holds that human beings have the capacity to know God through natural reason.

In conclusion, I show that Aristotle's cosmological proof, Avicenna's metaphysical proof, and Aquinas's five-way proof differ from each other because of their different epistemological principles, different metaphysical constructions, and particular theological intentions. Finally, I explain that philosophers and theologians who are dedicated to proving God's existence primarily aim not to prove God's existence, which they do not question at all, but to clarify the nature or essence of God, given the unification of God's existence and His essence. Through these proofs of God's existence, we finally arrive at the essence of God.

## 2. Aristotle's Cosmological Proof for the First Mover

To present Aristotle's cosmological proof, I begin by clarifying the metaphysical framework of Aristotle, within which the cosmological proof is unfolded. In *Metaph*. Γ1, Aristotle declares that "there is a science, (i.e., metaphysics) which investigates being qua being and the per se properties" (Ἔστιν ἐπιστήμη τις ἣ θεωρεῖ τὸ ὂν ἧ ὂν καὶ τὰ τούτῳ ὑπάρχοντα καθ' αὐτό, 1003a21-22). Next, Aristotle explains that although being is said in multiple senses, all of these senses point to one focal meaning, namely, substance (οὐσία).[1] Insofar as Aristotle's metaphysics studies being qua being in the sense of ousia, it is called ousiology. From the beginning, Aristotle recognizes two kinds of substances, sensible and intelligible.[2] A sensible substance is combined with matter and is movable; an intelligible substance is separated from matter and is immovable. Aristotle argues that if there were nothing other than sensible substance, then physics, which studies sensible substance, would be the first philosophy.[3] Since physics is not the first philosophy but theology is, the subject of theology, the intelligible substance, is a kind of substance that is different from sensible substance.[4] Moreover, Aristotle emphrasizes that there must be two kinds of substances, mainly because sensible and intelligible substances do not have a common origin.[5] There are two different kinds of substances: sensible and intelligible. Among the sensible substances, some are natural things that are generated and corrupted in the sublunar world, and others are heavenly bodies that are constantly moved throughout the universe. Therefore, Aristotle recognizes a total of three kinds of substances. In *Metaph*. Λ1, 1069a30-36, Aristotle summarizes the three kinds of substances by characterizing each of them as having two different and compatible attributes, one of them being sensible and perishable, another being sensible and imperishable, and the third being intelligible and immovable. Accordingly, Aristotle regards natural things, such as human beings, animals, and plants, as sensible and perishable substances; heavenly bodies as sensible and imperishable substances; and God as an intelligent and immovable substance.[6] Thus, Aristotle's ousiology examines a total of three kinds of substances: natural entities, heavenly bodies, and God. From these three kinds of substances, Aristotle further deduces two kinds of causes; accordingly, metaphysics shifts its form from ousiology to aitiology.[7] In my view, Aristotle's aitiology includes cosmological proof and the doctrine of the four causes. On the one hand, Aristotle grasps the four causes, i.e., material, formal, efficient, and final causes, by examining how a natural thing is generated and moved. On the other hand, he

introduces God as the First Mover by examining the ultimate efficient cause of the motion of the heavenly bodies. When we do not limit ourselves to the content of *Metaphysics* (especially the Book Γ) but look at Aristotle's works on physics such as *Physics, On the Heavens*, and *On Generation and Corruption* in a comprehensive way, the basic structure of Aristotle's metaphysics takes its full form: it appears either in the form of ousiology, which investigates three kinds of substances (natural things, heavenly bodies, and God), or in the form of aitiology, which investigates two kinds of causes (the four causes of the generation and motion of natural things, and the ultimate efficient cause of the motion of heavenly bodies).[8] Within the metaphysical framework of aitiology and ousiology, Aristotle constructs cosmological proof of God's existence, which deals with the causal relationship between two kinds of substances, that is, how the heavenly bodies are causally related to God.

Furthermore, I draw attention to the epistemological principle established by Aristotle, which forms the epistemological basis for his cosmological proof. According to Aristotle's epistemological principle, our cognitive activity should start from what is clearer and more knowable to us and eventually arrive at what is clearer and more knowable by nature (πέφυκε δὲ ἐκ τῶν γνωριμωτέρων ἡμῖν ἡ ὁδὸς καὶ σαφεστέρων ἐπὶ τὰ σαφέστερα τῇ φύσει καὶ γνωριμώτερα, *Phys.* A1, 184a16-18).[9] In other words, our cognitive activity should proceed from what is prior to us to what is prior by nature. For Aristotle, what is prior to us refers to what is prior to sensation, that is, sensible substances, which include natural things and heavenly bodies. To prove the existence of God, Aristotle starts not from natural things but from heavenly bodies because, in his view, only heavenly bodies in the universe have a direct relationship with God.

Based on what Aristotle proves in *Metaph.* Λ6-8 and *Phys.* Θ5, I reconstruct Aristotle's cosmological proof as follows. Having accepted the astronomical findings of his time, Aristotle believes that there are fifty-five heavenly bodies in the universe.[10] Aristotle also establishes an important principle, namely, that what is moved must be moved by something (ἐπεὶ δὲ τὸ κινούμενον ἀνάγκη ὑπό τινος κινεῖσθαι, *Metaph.* Λ8, 1073a26).[11] On this basis, Aristotle states that among these heavenly bodies, A is moved by B, B is moved by C, and C is moved by D. However, this would lead to an infinite regress, which Aristotle does not recognize. Because of the limited number of heavenly bodies, the chain of moving and being moved among them cannot enter infinite regression. In addition, Aristotle's universe does not allow for infinity because, in his view, the universe is finite, bounded, and cannot expand infinitely, in contrast to the Big Bang theory of recent scientific cosmologists, according to which the universe has no boundaries and is still expanding. In the case of an Aristotelian finite universe and a finite number of heavenly bodies, one can either recognize the last two heavenly bodies interacting or establish a heterogeneous first mover to end this infinite regression. Aristotle does not recognize the interaction between the heavenly bodies[12] but believes that there must be a first mover that moves the heavenly bodies without being moved by anything else.[13] Given the prohibition of infinite regression, Aristotle thus deduces God as the First Mover from the moved heavenly bodies.

In what follows, I examine the argument by arguing that to be valid, Aristotle's cosmological proof must satisfy two assumptions, one of which is the aforementioned prohibition of infinite regress, and the other is that the heavenly bodies must be purely material entities. The second assumption does not contradict the findings of modern science but was difficult for the ancients to accept; after all, the ancients believed that the heavenly bodies had souls, as seen in Alexander's (1891) commentary on *Metaph.* Λ8.[14] However, if the heavenly bodies were a combination of a cosmic soul and a cosmic body, they would be able to move on their own, just like a human being as a combination of a soul and a body, without the need for an external mover, and the entire cosmological proof would be invalidated.[15] Therefore, the heavenly bodies must be material entities that are generally incapable of actively moving themselves (οὐ γὰρ ἥ γε ὕλη κινήσει αὐτὴ ἑαυτήν, *Metaph.* Λ6, 1071b29-30). Just as a material entity, such as a table, cannot propel itself into motion, so too the heavenly bodies, as material entities, cannot propel themselves into motion;

thus, they need an external propeller or mover. This mover moves the heavenly bodies without being moved by anything else, so it is the Unmoved Mover (τοίνυν ἔστι τι ὃ οὐ κινούμενον κινεῖ, *Metaph*. Λ7, 1072a24-26). Since the Unmoved Mover is the ultimate efficient cause of the motion of the heavenly bodies, Aristotle calls Him the First Mover (τὸ πρῶτον κινοῦν).[16] Moreover, the First Mover can ultimately end the chain of moving and being moved only if it is essentially different from the moved heavenly bodies. If the First Mover contained matter that cannot move itself, then the First Mover would still need an efficient cause and thus fall back into an infinite regress. Therefore, the First Mover and the moved heavenly bodies must be heterogeneous, just as the person who pushes a chain of dominoes is essentially different from the dominoes that are pushed. Since the heavenly bodies are purely material entities, the First Mover must be a purely intelligent entity, that is, God.[17] Through his cosmological proof, Aristotle not only proves that God is the First Mover but also concludes that God, as the First Mover, must be a purely intelligent substance, a conclusion consistent with his original presupposition of God. According to Aristotle, God, as an intelligent substance, actualizes itself by thinking of itself eternally and continuously (ἔστιν ἡ νόησις νοήσεως νόησις, *Metaph*. Λ9, 1074b33-35).[18] Through eternal and continuous self-realization (actus purus, ἡ ἁπλῆ καὶ κατ᾽ ἐνέργειαν, *Metaph*. Λ7, 1072a31-32), God directly drives the motion of the heavenly bodies and indirectly influences the birth and death of natural things, thus playing the role of the ultimate efficient cause of the universe and the sublunar world.[19]

In sum, Aristotle constructs the cosmological proof that proceeds from the moved heavenly bodies to God as the First Mover not only because he follows the epistemological principle that proceeds from the sensible to the intelligible but more fundamentally because causality, specifically the causal relationship in the sense of moving and being moved between God and the heavenly bodies, plays a crucial role in Aristotle's metaphysical framework of ousiology and aitiology. Crucially, Aristotle's cosmological proof establishes a paradigm for deducing God by argument, inferring cause from effect. Consequently, the later Peripatetics, in discussing related issues, had to target this Aristotelian paradigm, as seen in Avicenna's criticism of it and Aquinas's adherence to it.

### 3. Avicenna's Metaphysical Proof for Necessary Existence in Itself

Regarding Avicenna's proof of God's existence, scholars have mainly debated two related questions: Where can this proof be found and what is its nature? Regarding the first question, I argue that Avicenna demonstrates God's existence in the *Metaphysics of the Healing* (al-Šifā: Ilāhīyyāt, abbreviated as *Ilāhīyyāt*), following Menn (2013), who claims that there is proof of God's existence "briefly in (*Ilāhīyyāt*) I.6-7 and more fully in Book VIII" (2013, p. 149, n. 14).[20] In my view, a few relevant passages scattered throughout *Ilāhīyyāt* I.2, I.5-7, and VIII.1-3 constitute the entire proof, with each of these passages playing a different role in the construction of the proof. Regarding the nature of Avicenna's proof,[21] I agree with Mayer, who sees Avicenna's proof as both metaphysical (in his words, "ontological")[22] and cosmological (2001, pp. 35–39). While Mayer focuses on the proof Avicenna proposes in *Najāt* and *Ishārāt*, I argue that in *Ilāhīyyāt* (I.2, I.5-7, VIII.1-3), Avicenna uses syllogism to prove God's existence by deducing God, who has a special existence, from the universal concept of existence. In my interpretation, Avicenna's proof proceeds from the universal to the particular, so it does not follow the Aristotelian pattern of cosmological proof, which proceeds from effect to cause. Thus, Avicenna's proof is not cosmological but rather metaphysical. In what follows, I show how Avicenna provides a metaphysical proof for the existence of God in *Ilāhīyyāt*, beginning with Avicenna's critique of Aristotle's cosmological proof.

In the commentary on Aristotle's *Metaph*. Λ7, Avicenna criticizes Aristotle's cosmological proof by claiming that even if the argument were valid, it would not prove a cause for the existence of the heavenly bodies but merely identify God, whom Avicenna calls, in this context, "the First Real", "the One", "the Real", or "the First Principle", with the efficient cause of the motion of the heavenly bodies. From Aristotle's identification of God with

the ultimate efficient cause of the motion of the heavenly bodies, according to Avicenna, Aristotle cannot necessarily take this efficient cause (mabdaʾ li-haraka = causa efficiens) to be an existential cause of the heavenly bodies (mabdaʾ li-ḏuwāt/mabdaʾ li-mauǧūd = causa essendi).[23] The existential cause is related to the question of how the heavenly bodies come into being, while the efficient cause is related to the question of how the heavenly bodies are constantly moved. Efficient cause and existential cause are different things. Thus, in Avicenna's view, Aristotle's cosmological proof at most proves that there is a heterogeneous First Mover as the efficient cause beyond the heavenly bodies, but it does not prove God as the existential cause of the heavenly bodies and all other things. Moreover, Avicenna might criticize that the cosmological proof, which proceeds from effect to cause, lacks necessity. For example, if we see water on the road, it does not necessarily follow that this is the result of rain from the sky because there are many other causes that could lead to the same result. Similarly, even if a causal relationship of moving and being moved is recognized between God and the heavenly bodies, it is not necessary to infer God from the heavenly bodies, and the causal relationship is only somewhat contingent.[24] Gutas alludes to this criticism by quoting a passage from an older contemporary of Avicenna named Abū-Sulaymān as-Sijistānī as follows (2014, pp. 297–98):

The most appropriate inquiry concerning the first mover is that in which discussion of Physical Theory is combined with Metaphysical theory. [. . .] Let us then state in what way this is so. Inquiry concerning the conjunction of effects with causes has two aspects; the first, insofar as it ascends through the connection of one to the other to their causes; and the second, insofar as the permeation of the power of the cause in its effects [is considered]. Inquiry in the first mode belongs to the Physicist, and in the second mode to the Metaphysician [i.e., qua student of Universe Science]. There exists also a third mode unconcerned with relation, namely, inquiry into the essence apart from affinities and relationships. The discussion belongs to the Theologian.

In this passage, Abū-Sulaymān mentions three ways to reach God. One is a physical approach, another is a metaphysical approach, and the third is a theological approach. In Abū-Sulaymān's view, with which Avicenna agrees, Aristotle's cosmological proof takes the physical approach of inferring cause from effect. While the physical approach of inferring cause from effect lacks necessity, as mentioned above, the metaphysical approach of inferring effect from cause has necessity. In my opinion, Avicenna adopts a metaphysical approach with some modifications. He modifies the metaphysical approach of inferring effect from cause proposed by Abū-Sulaymān into his own version of inferring the particular from the universal, which also guarantees the necessity of deduction. By emphasizing necessity, Avicenna favors the metaphysical approach over the physical approach, thereby replacing Aristotle's cosmological proof with a metaphysical proof. While the physical and metaphysical approaches deal with the relationship between cause and effect, the theological approach stands apart from causal relationships and affinities and instead requires an exploration of the nature of God Himself. The theological approach described here refers to negative theology, according to which God remains beyond causal relations and can be expressed only in negative terms. I agree with Gutas that Avicenna abandons the Aristotelian physical approach in proving the existence of God. However, I disagree with his claim that Avicenna "establishes the First Real by means of the theological way" (Gutas 2014, p. 299) because the latter does not take a theological approach in the sense of negative theology but rather a metaphysical approach, which necessarily derives God, the particular Supreme Being, from the universal being.

Moreover, Gutas aptly notes that "Avicenna's criticism of Aristotle's proof of the First Mover by means of motion and his concomitant theory of Necessary Existence are among the most important aspects of this metaphysical system" (2014, pp. 299–300). In this way, Gutas suggests that Avicenna establishes a metaphysical proof while criticizing Aristotle's cosmological proof not only because Aristotle's arguments lack necessity but also because Avicenna reconstructs the entire metaphysical framework to such an extent that God within it must be explored in a different way. I elaborate on this point below.

In the reconstruction of metaphysics, Avicenna made a twofold effort to redefine the subject and method of this discipline. As Bertolacci has clearly and aptly noted, Avicenna distinguishes between the subject (mawḍūʿ = ὑποκείμενον) and the goal (ǧaraḍ = τέλος) of metaphysics; on this basis, he argues that God cannot be the subject of metaphysics, only its goal (2006, pp. 111–31). In the *Analytics Posterior*, Aristotle establishes the scientific principle that a discipline cannot prove its own subject and that the subject must be posited as something self-evident to that disciple, and he applies it to scientific disciplines such as arithmetic and geometry.[25] Arithmetic cannot by itself prove the existence of numbers and their essential properties, such as oddness and evenness. Geometry cannot by itself prove the existence and properties of lines, surfaces, and solids; moreover, axioms must be established within geometry and used as self-evident premises. Avicenna applies this scientific principle of Aristotle to metaphysics. Accordingly, metaphysics cannot by itself prove the existence of its subject but must take a self-evident subject as the starting point. God's existence is not self-evident but must be proven, so God cannot be regarded as the subject of metaphysics, only as its goal. Although God is not self-evident, there are some things that are self-evident, which Avicenna calls the Firsts. To redefine "first", which Aristotle defines as a first-class being, i.e., a purely intelligent being, or the first mover, i.e., the ultimate efficient cause, Avicenna modifies Aristotle's epistemological principle. According to Aristotle, our cognitive activity should proceed from what is prior to us to what is prior by nature. By what is prior to us, Aristotle means what is prior to sensation, i.e., the sensible and particular, while Avicenna means what is prior to human intellect, i.e., the intelligible and universal, by asserting in *Ilāhīyyāt* I.5 that "the concepts (maʿnā) of 'existent', 'thing', and 'necessary' are impressed upon the soul as a primary impression; this impression does not need to be obtained from better-known things."[26] Avicenna reinterprets Aristotle's epistemological principle by identifying the firsts with intelligible and universal concepts, such as existent (mawjūd = ens), thing (šayʾ = res), one (wāḥid = unum), and necessity (ḍarūrī = necessitas).[27] These universal concepts are the first encountered by human intellect and are therefore self-evident; thus, they should be the subject of metaphysics. Metaphysics should investigate universal concepts in general and the existent in particular, as Avicenna emphasizes in *Ilāhīyyāt* I.2 as follows.

The primary subject of this science (i.e., metaphysics) is the existent insofar as it is an existent, and what is (also)sought in this science are the things that accompany the existent insofar as it is an existent unconditionally (*Ilāhīyyāt* I.2, p. 10, lines 3–6, translation by Marmura with slight modification).

In order to explain exactly the aforementioned things that unconditionally accompany the existent qua existent, Avicenna distinguishes two cases. In the first case, just as a genus is divided into species, existent, which is equivalent to being in this context, can be divided into categories such as substance, quantity, and quality (*Ilāhīyyāt* I.2, p. 10, lines 7–8). The categorical division of being into substance, quantity, and quality is accompanied by the specification of being physical, mathematical, or moral. Although categories accompany being, being is no longer universal insofar as it is unconditioned being. Thus, categories are not what metaphysics should be looking for. Rather, metaphysics should look for properties per se that do not specify being as physical, mathematical, or moral, as Avicenna mentions below.

Some of these things (sought in metaphysics) are similar to the properties per se (al-ʿawāriḍ al-ḫāṣa), such as the one and the many, potency and actualization, the universal and the particular, and the possible and the necessary. For the existent, in accepting these properties and in being prepared for them, does not need to be specified as physical, mathematical, moral, or any other (*Ilāhīyyāt* I.2, p. 10, lines 11–16, translation by Marmura with slight modification).

Thus, according to Avicenna, metaphysics should investigate the existent qua existent and the properties per se that unconditionally accompany the existent. These disjunctive essential properties include one–many, potency–actualization, universal–particular, and possible–necessary. Avicenna is obviously constructing metaphysics as an ontology, i.e.,

metaphysica generalis.[28] Avicenna's dictum seems to bear a great resemblance to Aristotle's statement that "there is a science (i.e., metaphysics) which investigates being qua being and the properties per se" (Ἔστιν ἐπιστήμη τις ἣ θεωρεῖ τὸ ὂν ᾗ ὂν καὶ τὰ τούτῳ ὑπάρχοντα καθ' αὑτό, *Metaph*. Γ1, 1003a21-22). Nevertheless, in *Metaph*. Γ1-3, Aristotle clearly relates being qua being with principle or substance. Moreover, in *Metaph*. Z13, Aristotle explicitly rejects the universal as substance (ἔοικε γὰρ ἀδύνατον εἶναι οὐσίαν εἶναι ὁτιοῦν τῶν καθόλου λεγομένων, 1038b8-9), thereby criticizing the two kinds of universals, i.e., Platonic ideas and Pythagorean numbers, as substances in *Metaph*. Z14 and Z16 and in the last two books, M and N. Therefore, it is impossible for Aristotle to investigate being qua being in the universal, unconditioned sense or to establish a metaphysica generalis. The establishment of a metaphysica generalis is the result of Avicenna's significant reworking and modification of Aristotle's metaphysics, which in turn influences Aquinas's view of Aristotelian metaphysics and his own construction of metaphysics.[29]

In order to reconstruct metaphysics as a universal science, Avicenna also redefines the method of this discipline by adopting and modifying Aristotle's scientific method, syllogism. Following Aristotle, who uses the syllogistic method to construct scientific disciplines such as arithmetic, geometry, and astronomy in the *Analytics Posterior*, Avicenna shows that metaphysics, like other scientific disciplines, should be constructed using syllogism.[30] As mentioned above, Avicenna asserts in *Ilāhīyyāt* I.5 (p. 22, lines 19–20) that the concepts of "existent", "thing", and "necessary" are impressed upon the soul as a primary impression. This impression is primary in the sense that these universal, self-evident concepts cannot be obtained through less universal, less self-evident concepts. Conversely, less universal concepts should be inferred from universal concepts, just as a particular proposition is inferred from the universal proposition that functions as the major premise (which Avicenna calls the "first principle") of the demonstration and that is true in itself (*Ilāhīyyāt* I.5, p. 22, lines 20–24). Just as the particular true proposition "Socrates is mortal" is validly deduced from the universally true proposition "All men are mortal" (through the transition of the middle term "Socrates is a man"), so too can a particular concept be deduced from a universal concept. On this basis, I propose that Avicenna uses the modified syllogism to provide a metaphysical proof of God's existence.

Avicenna begins with the existent qua existent, i.e., the universal, self-evident concept of the existent (*Ilāhīyyāt* I.2, I.5). He then mentions that "the things that come into existence can be dichotomously divided in the mind" into the necessary existent and the possible existent (*Ilāhīyyāt* I.6, pp. 29–30). When the possible existent is considered by itself, its existence is neither necessary nor impossible (*Ilāhīyyāt* I.6, p. 30, lines 1–5). When the necessary existent is considered by itself, its existence is necessary (*Ilāhīyyāt* I.6, p. 30, lines 5–6); in other words, its nonexistence is impossible and inconceivable. More precisely, the dichotomy appears to be possible existent in itself and necessary existent in itself. Avicenna then argues that the necessary existent in itself differs from the necessary existent through another because the existence of the latter has a cause and depends on the cause and that of the former has no cause, being independent of anything else (*Ilāhīyyāt* I.6, p. 30, lines 7–31). In *Ilāhīyyāt* I.6, Avicenna actually divides existence into three kinds, namely, the possible existent in itself, the necessary existent through another, and the necessary existent in itself. Many scholars have drawn attention to this crucial threefold division of existence and given different interpretations. On the basis of careful textual analysis, Bertolacci has reconstructed this division as a diairesis, so that existence is divided into the pair necessary–possible, and these two, i.e., necessary existent and possible existent, are subdivided into the pair in itself–by another, respectively.[31] On the other hand, Wisnovsky describes Avicenna's division as the matrix of distinctions, in which existence is divided into pairs such as necessary–possible, in itself–through another, and uncaused–caused (Wisnovsky 2003, pp. 197–263; De Haan 2020, pp. 150–79). Following Wisnovsky, I propose that Avicenna constructs a matrix in such a way that he divides existence from different perspectives, differentiating existence in parallel into necessary–possible with respect to the mode of being, in itself–through another with respect to the origin of necessity, uncaused–caused

with respect to whether the existent has a cause, and simplicity–multiplicity with respect to whether the existent is simple or composed (Wisnovsky 2003, pp. 197–98, 248, 260–63; De Haan 2020, p. 172). Since the distinctions that Avicenna constructs are not ordered in a vertical, hierarchical way but are arranged in parallel, Avicenna does not perform a diairesis (as Bertolacci assumes) but rather constructs a matrix (as Wisnovsky identifies). On this basis, I suggest that Avicenna does not stop at the matrix of distinctions but rather goes on to construct a cross-division.[32] In my opinion, Avicenna divides existence in parallel into necessary–possible and in itself–through another. These two pairs are cross-combined with each other, thus establishing four combinations that characterize four kinds of existents, namely, necessary existent in itself, necessary existent through another, possible existent in itself, and possible existent through another. The possible existent through another, that is, the existent that derives its possibility from another, is not actually possible in reality, so Avicenna omits this kind of existent and focuses on the first three kinds of existents. Avicenna further equates possible existents in themselves with necessary existents through another because possible existents can only come into existence through the other existents of a higher rank, i.e., necessary existents. In other words, beings exist by themselves only in a possible way, but they exist in reality, so their actual existence must be achieved through a necessary other. Among the necessary existents through another, one derives its necessity from another, this other derives its necessity from another, and in this way, the process goes on for infinity. However, the chain of necessary existents deriving their necessity from another cannot go on for infinity, so Avicenna concludes that to end this chain, there must be the necessary existent that derives its necessity from itself, that is, God (*Ilāhīyyāt* I.6, p. 31, line 27-p. 32, line 6). God, as necessary existent in itself, is essentially different from necessary existent through another (*Ilāhīyyāt* I.7, p. 35, lines 14–20). Whereas the necessary existent through another is caused and composite, God is uncaused and simple, completely isolated from potentiality and possibility (*Ilāhīyyāt* I.7, p. 38, lines 7–21).

As Davidson (1987, pp. 298–99) and Bertolacci (2006, p. 224) have rightly pointed out, Avicenna does not construct a metaphysical proof that proceeds from cause to effect because God is not self-evident and cannot be taken as the starting point of the proof. Nevertheless, nothing prevents Avicenna from constructing a metaphysical proof that proceeds from the universal to the particular. The proof of God's existence is the process by which humankind comes to know God. Humankind can reach God as a particular existent only through the universal existent, which, according to Avicenna, is self-evident to the human intellect. The crucial point is that the process of knowing God does not compromise the universality of God Himself. Thus, I propose that Avicenna constructs a metaphysical proof using a syllogism in which the universal, self-evident concept of existence is established as the major premise (*Ilāhīyyāt* I.2, I.5). To establish the middle term, Avicenna divides existence into three kinds and reduces them to two kinds by equating the possible existent in itself with the necessary existent through another (*Ilāhīyyāt* I.6). Since the chain of necessary existents through another cannot go on for infinity, necessary existents through another must ultimately be attributed to the necessary existent in itself (*Ilāhīyyāt* I.6-7). Thus, Avicenna concludes that God exists by syllogistically inferring God as the special existent, i.e., the necessary existent in itself, from the universal existent, i.e., the existent qua existent. Avicenna anticipates the conclusion in *Ilāhīyyāt* I.6-7, which is fully articulated in VIII.1-3. Avicenna's proof follows the syllogistic pattern of inferring the particular from the universal, which is inconsistent with Aristotle's cosmological proof of inferring cause from effect, so that Avicenna's proof as a whole is not cosmological but rather metaphysical. Nevertheless, Avicenna's metaphysical proof has elements of cosmological proof since Avicenna establishes a certain causal relationship between two kinds of necessary existents and rejects infinite progression (Davidson 1987, p. 180).

In summary, Avicenna constructs a metaphysical proof while directing his criticism at Aristotle's cosmological proof because he modifies Aristotle's epistemological principle of proceeding from the sensible to the intelligible into the epistemological principle of deduction from the universal to the particular. More fundamentally, Avicenna rejects Aristotle's

way of constructing metaphysics as ousiology and aitiology and reconstructs this discipline as ontology, i.e., metaphysica generalis.[33] To this end, he defines the subject of metaphysics as universal existence and establishes the method of this discipline as syllogism. Crucially, Avicenna's twofold modification of metaphysics, that is, the redefinition of the subject and the method, are interrelated to form his metaphysical proof of God's existence. Avicenna constructs metaphysics by means of syllogism, so that metaphysicians should start from the subject of metaphysics, i.e., universal existence, and arrive at the goal of this discipline, i.e., God's existence.[34] Accordingly, Avicenna establishes a metaphysical proof in a similar syllogistic manner so that one proceeds from universal existence to special existence, i.e., God as the necessary existent in itself. Thus, Avicenna's establishment of the metaphysical proof of God's existence is closely related to his reconstruction of metaphysics.

### 4. Thomas Aquinas's Cosmological Proof for the Existence of God

As analyzed, Avicenna criticizes Aristotle's cosmological proof because the Aristotelian way of inferring cause from effect lacks necessity. Thus, Avicenna constructs a metaphysical proof using syllogism, which necessarily infers the particular from the universal. Accordingly, Avicenna constructs a metaphysical proof in such a syllogistic way that he deduces God as the special existent from the universal existent. Although Aquinas does not explicitly mention Avicenna in his proof of God's existence, it is possible to reproduce Aquinas's criticism of Avicenna based on his philosophical principles. Aquinas might think that Avicenna's syllogistic method, while having necessity, derives only the particular concept of God, i.e., "the necessary existent in itself", from the universal concept of "existent" and does not prove the real existence of God. Aquinas believes that God's existence cannot be inferred from any concept but rather from only some other existence; therefore, he abandons Avicenna's metaphysical proof by the concept of the existent and returns to Aristotle's cosmological proof by causality. Aquinas thus follows the Aristotelian pattern of inferring cause from effect in the five-way proof. In what follows, I examine the five-way proof presented in *Summa Theologiae* I, q. 2, a. 3, focusing mainly on the first three ways, which are essentially associated with Aristotle's cosmological proof and Avicenna's metaphysical proof.

The first way is based on motion. Although it is closest to Aristotle's cosmological proof, Aquinas's first way does not start from the motion of heavenly bodies in the universe but rather from the motion of natural bodies in the sublunar world. I reconstruct the argument in the following four steps. (1) By means of sense perception, we observe that some things are moved in the sublunar world. (2) Whatever is moved is moved by something else. (2.1) To move is to bring something from potentiality to actuality; moreover, a thing can be brought from potentiality to actuality only by another thing that is in actuality. For example, fire, which is actually hot, makes wood, which is potentially hot, actually hot, so fire moves and changes wood. (2.2) However, it is impossible for the same thing to be both actual and potential at the same time and in the same way, as what is actually hot cannot be potentially hot at the same time but can be potentially cold at the same time. (2.3) It is therefore impossible for something to be both mover and moved in the same respect and in the same way or for it to move itself. Therefore, whatever is moved must be moved by something else. (3) Furthermore, this something else, if it is moved, must itself be moved by something else, and that something else by yet another thing. In this way, the chain of moving and being moved would go on for infinity. However, this chain cannot go on for infinity because if there were no first mover, then there would be no other movers. The second movers can move something only if they are moved by a first mover, just as a stick can move only if it is moved by a hand. (4) Therefore, it is necessary to have as the first mover something that is not moved by anything, that is, God.

Regarding the first way, scholars debate whether Aquinas uses movetur in the intransitive sense of "in motion" or in the passive sense of "being moved".[35] I prefer the latter reading and believe that Aquinas consistently uses movetur in the passive sense. For Aquinas states that "whatever is moved must be moved by something else" (omne ergo

quod movetur, oportet ab alio moveri) by imitating the principle established by Aristotle in the cosmological proof that "whatever is moved must be moved by something" (ἐπεὶ δὲ τὸ κινούμενον ἀνάγκη ὑπό τινος κινεῖσθαι, *Metaph.* Λ8, 1073a26). From the Greek original, it is clear that movetur, like κινεῖσθαι, is used in the passive sense.[36] After asserting that things are moved in the sublunar world (1), Aquinas turns to explaining two premises of the argument that play a crucial role in constructing the first way. Aquinas is so aware of the importance of these two premises that he provides a defense of them in *Summa Contra Gentiles* I, c.13. The first premise is that whatever is moved must be moved by something else, as stated in (2) and proven by (2.1)–(2.3). The second premise is the prohibition of infinite regress, stated in (3). As scholars have noted, there are many problems with these premises, which I will not repeat. What I am questioning is the first premise, in which one cannot derive (2.3) from (2.2). In my view, Aquinas confuses two sets of concepts here by substituting the non-coexistence of potentiality and actuality (2.2) for the non-coexistence of moving and being moved (2.3). It is true that potentiality and actuality cannot coexist simultaneously, but moving and being moved can coexist simultaneously; e.g., according to Aristotle, all living things (such as plants, animals, and humans) and God can move themselves (Shields and Pasnau 2016, p. 110).[37] Although one cannot derive the non-coexistence of moving and being moved from the non-coexistence of potentiality and actuality, Aquinas is correct in asserting that it is impossible for something to be both mover and moved in the same respect and in the same way. He offers a valid defense of this assertion in *SCG* I, c.13. Aquinas argues that natural things seem to move themselves so that plants grow, animals walk, and humans act, but this kind of self-motion is merely motion per accidens, not motion per se. Regarding motion per se, things should be considered as a whole in order to move themselves as a whole; natural things, whether plants, animals, or human beings, are not considered as a whole in order to move the whole but as one part in order to move another part, just as the vegetative, animal, and human souls move their respective bodies. Aquinas thus concludes that natural things, which Aristotle regards as causa sui, do not actually move themselves but only appear to do so.[38] In this context, Aquinas strongly opposes causa sui; this is actually a major reworking of Aristotle's philosophy, since the former defines natural things as those that move themselves. According to Aristotle, a natural thing moves itself because it has its efficient cause in itself, whereas an artificial thing cannot move itself because it has its efficient cause in another. By distinguishing between the internal efficient cause and the external efficient cause, Aristotle seeks to distinguish between a natural thing and an artificial thing.[39] For Aquinas, however, the distinction between natural and artificial things is not important; he is concerned with the distinction between creatures and the Creator. Accordingly, Aquinas places the efficient cause of a creature, whether natural or artificial, outside itself, thereby interpreting the efficient cause as an external cause.[40] Aquinas's interpretation of the efficient cause is consistent with his theological intention and with Catholic doctrine, such that the ultimate efficient cause of a thing cannot be the thing itself but must be an external existence, i.e., God. Thus, a thing in the sublunar world, whatever it is, insofar as it is moved, must be moved by another and that again by another. However, Aquinas, following Aristotle, forbids infinite regression, so the chain of moving and being moved must end with a first mover who moves all others without being moved by anything else. Consistent with Aristotle's prohibition of infinite regression, Aquinas ultimately deduces God as the First Mover from the moved things, just as Aristotle does in his cosmological proof.

The second way is based on the hierarchy of efficient causes. I reconstruct the argument as follows. (1) By means of sense perception, we observe that there is a hierarchical order of efficient causes among sensible things. (2) It is impossible for a thing to be the efficient cause of itself. To be the efficient cause of itself, the thing must exist before itself, which is impossible. (3) It is impossible for efficient causes that are hierarchically ordered to go on for infinity. For in any ordered series of efficient causes, the first member causes the intermediate member, which is either one or many, and the intermediate member causes the final member. If a series of efficient causes could go on for infinity, that is, if the first

cause could be eliminated, then the intermediate and final causes would be eliminated, and in the case of the elimination of causes, the effect would also be eliminated. This cannot happen, and so it is wrong. (4) Therefore, it is necessary to posit something as the first efficient cause, hence God.

After assuming that sensible things have multiple efficient causes that are hierarchically ordered (1), Aquinas argues the following two points, which are consistent with the two premises of the first way. First (2), Aquinas claims that whatever is efficiently caused must be caused by something else. As Wippel (2000, p. 460) has correctly noted, this point echoes the aforementioned principle against causa sui, so that whatever is moved must be moved by something else. Then (3), Aquinas rejects the infinite regression of the series of efficient causes, just as he prohibits the infinite regression of moving and being moved. To illustrate this point, Gilson and Kenny use a natural phenomenon as an example. They mention that in nature, the son is begotten by the father, the father by the grandfather, and so on. However, as Gilson has aptly pointed out, if the series of efficient causes could be limited to the natural world, it could go on for infinity.[41] Thus, I follow him in assuming that by the series of efficient causes, Aquinas does not mean a series such as a father, grandfather, and so on; rather, he means the efficient causes that are arranged hierarchically, such as father, sun, and God. God, as the first efficient cause, moves the heavenly bodies in general and the sun in particular. The sun, in turn, affects the generation and decay of plants and animals, as well as human beings in the sublunar world; for example, the son is begotten by his father when the sun is at perihelion. Therefore, Aquinas places the hierarchical structure among the efficient causes and introduces the first efficient cause by opposing causa sui and prohibiting infinite regression. In this vein, Aquinas follows in Aristotle's footsteps and constructs the second way similarly to the first. While the first way focuses on God as the First Mover, i.e., the ultimate efficient cause of motion, the second way emphasizes God as the ultimate efficient cause of existence. The second way allows Aquinas to respond to Avicenna's criticism of Aristotle's cosmological proof that God is not only the ultimate efficient cause of motion but also of existence.

The third way is based on the possible and the necessary. I reconstruct the argument as follows. (1) In the sublunar world, we find some things that have the possibility of being and not being since we observe that these things are generated and corrupted. Therefore, there is a kind of being that has the possibility of being and not being, that is, the possible being. (2) However, it is impossible for all things to be like that because the possible being that has the possibility of not being does not exist at some time (quandoque). (2.1) If all things were possible beings with the possibility of not being, there would have been nothing at some time in the past (aliquando). (2.2) If that were the case, there would still be nothing now (nunc) because what is not can only begin to be through something that is. If there were nothing, it would be impossible for anything to begin to be, so there would be nothing even now. This does not happen, so it is wrong. Therefore, not all things are possible beings, but there must be necessary beings among things. (3) Among these necessary beings, some have the cause of their necessity through another, and the other has the cause of necessity in itself. Just as the chain of efficient causes cannot go on for infinity, the chain of necessary beings having the cause of their necessity through another cannot go on for infinity. (4) In order to terminate this chain, it is necessary to posit something that is necessary per se, that is, the necessary being in itself. The necessary being in itself has the cause of necessity not through another but in itself, thus being the cause of other necessary beings, that is, God.

Aquinas (1) begins with possible existents, which have the possibility to both be and not to be, as seen in their generation and decay. He then uses reductio ad absurdum to prove that in addition to possible existents, there must be necessary existents. Regarding the reductio ad absurdum argument, Wippel notes that scholars hotly debate whether Aquinas uses the two terms quandoque and aliquando to refer to the nonexistent at some time in the future or at some time in the past (Wippel 2000, p. 464, n. 57). I agree with Wippel that aliquando refers to a nonexistent at some time in the past. While Wippel

sees aliquando and quandoque as equivalent, I see them as different. Therefore, I offer an alternative interpretation, taking (2) as the general expression according to which it is impossible for all things to be possible existents, because there is some time (quandoque) at which each possible existent does not exist. Quandoque is not qualified, so it can refer to being nonexistent at some time in the past or nonexistent now; in this case, Aquinas then deals with the two possibilities. (2.1) If all things were possible existents, then at some time in the past (aliquando), there would have been nothing; (2.2) if this were the case, then, consequently, there would still be nothing now (nunc). However, this is not the case. There are things now (nunc), and in the past (aliquando), there was no time when there was nothing, so there is no time when there is or was nothing (quandoque). Thus, it is impossible for all beings to be possible existents, and there must be necessary existents. Wippel's interpretation is sound without distinguishing between quandoque and aliquando. I make this distinction not only because these two terms are different but mainly because, in the case of Wippel's interpretation, it is not clear why Aquinas would express the same meaning in two different ways, (2) and (2.1). To avoid duplication, I emphasize the distinction between quandoque and aliquando. Moreover, there is another problem with the reductio ad absurdum argument. From the antecedent "all things are possible existents", one cannot draw the consequence "there was nothing at some time in the past, and there would be nothing now", but only the consequence "there could be nothing at some time in the past, and there could be nothing now".[42] The problem is that Aquinas unilaterally emphasizes that possible existents have the possibility of not being, while deliberately ignoring the fact that possible existents also have the possibility of being. However, whether he emphasizes that possible existents have the possibility of being or the possibility of not being, Aquinas can still demonstrate beyond any doubt that possible existents can begin to exist only through existents of a higher rank, i.e., necessary existents. These necessary existents derive their necessity from another, so Kenny (1969, p. 48) correctly equates them with heavenly bodies, human souls, and angels. Since the chain of necessary existents deriving their necessity from another cannot go on for infinity, there must be a necessary existent that derives its necessity not from another but only from itself, that is, the necessary existent in itself, or God. Ultimately, Aquinas not only proves the existence of God but also characterizes God as the necessary existent in itself.

Although Aquinas's third way seems similar to Avicenna's metaphysical proof, I argue that it bears much more resemblance to Aristotle's cosmological proof. Avicenna uses syllogism to construct a metaphysical proof, which proceeds from universal existent to special existent, that is, from the universal concept of existent to God as the necessary existent in itself. In contrast, Aquinas does not use syllogism to establish the third way, which proceeds from one special existent to the other special existent, that is, from the necessary existent through another to the necessary existent in itself. Unlike Avicenna, who considers possibility and necessity to be two modes of being existent, and thus identifies two kinds of existents—i.e., possible existent and necessary existent—at the same time (*Ilāhīyyāt* I.6), Aquinas, after acknowledging possible existents, uses the reductio ad absurdum argument to prove that necessary existents, which derive their necessity from another, must exist. It is only in the final step of the third way, similar to Avicenna, that Aquinas introduces the necessary existent in itself and identifies it with God since the chain of necessary existents through another cannot go on for infinity and must have a terminus. Aquinas thus follows the Aristotelian pattern of inferring one special existent from other special existents. Just as in the cosmological proof Aristotle deduces God as the First Mover from the moved heavenly bodies, so in the third way, Aquinas deduces God as the necessary existent in itself from the necessary existents through another, such as the heavenly bodies, human souls, and angels.

The fourth way is based on the hierarchy of things. According to Aquinas, we find that among things, some are more or less good, noble, or the like that is signified by F, than others. Things are said to be more or less F according to the extent to which they approach something that is most F. Aquinas then claims on the authority of Aristotle that

something that is the best, truest, and noblest of all things is consequently the Supreme Being (maxime ens).[43] Aquinas also uses the example of fire. Just as fire, as the hottest thing, is the cause of all other hot things, so something that is most F is the cause of all those things that are more or less F. Aquinas thus concludes that there is something that is the cause of all being, goodness, and perfection, that is, God. As Kenny (1969, p. 71) aptly points out, all Thomists, while taking different positions on the interpretation of the fourth way, agree that on this point, "for better or worse, Aquinas comes closest to Platonism".[44] I agree with Kenny that Aquinas is ostensibly appealing to Aristotle here, but in fact, he is following Platonism. According to Platonism, all things are ordered in a hierarchical way, such that intelligible things transcend sensible things, and among intelligible things, which are Platonic ideas, ideas of a higher rank (which Plato calls the highest kinds, such as rest, motion, identity, difference, being and not-being)[45] transcend ideas of a lower rank (the ideas of artifacts or natural things).[46] At the top of this hierarchy is the Supreme Idea, which Plato calls the Idea of the Good (*Resp.* 507b5-7). Through the allegory of the sun, Plato claims that the Idea of the Good transcends all other ideas by bringing them into being and making them knowable and is thus the cause of all other beings, being both the cause of existence and the cause of knowledge (*Resp.* 509b2-10). In my view, what Aquinas means by God here is similar to the Platonic idea of the Good. Like the Platonic idea of the Good, Aquinas's God is the cause of all being, all goodness, and all perfection and thus is the Supreme Being, the best and the most perfect. Following Platonism, Aquinas thus illuminates a way of characterizing God, that is, via eminentiae (Wippel 2000, p. 474). Accordingly, the superlative form of the adjective referring to a certain quality F is used to describe God as the most F. Therefore, in my view, Kenny is very careful and precise in saying that Aquinas is closest to Platonism here, while excluding Neoplatonism, even though many Neoplatonists do not admit any difference between the two. Wippel (2000, p. 469) also notes that in the fourth way, Aquinas is inspired by Platonic and Neoplatonic sources without distinguishing between them. In the present context, there is a crucial difference between Neoplatonism and Platonism. Following Platonism, Aquinas here presents God in an absolutely positive way (via eminentiae), describing God as the Supreme Being, who is thus accessible to the human intellect. According to Neoplatonism, however, God cannot be portrayed in any positive way but should rather be characterized in an absolutely negative way (via negationis),[47] because God, like the Idea of the Good interpreted by Plotinus in the classical Neoplatonic way, transcends both being and intellect (τἀγαθὸν καὶ τὸ ἐπέκεινα νοῦ καὶ ἐπέκεινα οὐσίας, *Enneades* V. 1.8.6-8),[48] such that God is not being and is not accessible to the human intellect, thus being unthinkable and unsayable for humans. Neoplatonism lays the foundation for negative theology, while Platonism contributes to the establishment of natural theology. Therefore, what Aquinas follows here is not Neoplatonism but rather Platonism.[49]

The fifth way is based on the governance of things. We observe that natural things always or often act in the same way to achieve their goal. This observation shows that things do not achieve their goal by chance but rather because they intend to do so. Natural things that lack consciousness tend toward a goal only when they are directed by something with consciousness and intelligence. Therefore, there is something intelligent that directs all natural things toward a goal, and that something is God. Obviously, this proof is based on natural teleology, according to which natural things act in an orderly fashion and pursue an ideal. Natural things, lacking intelligence, can only pursue the ideal if they are directed by a supernatural, intelligent being, i.e., God.

Among the past and present Thomists who have offered different interpretations of the five ways, some have drawn attention to a single way and some to multiple but not all ways (MacDonald 1991; Shields and Pasnau 2016),[50] and still others have interpreted the five ways as a whole (Gilson 1957, 2002; Kenny 1969; Martin 1997; Wippel 2000; Pawl 2012; De Haan 2013). Wippel summarizes the unified interpretation into three approaches: some search for historical sources that Aquinas might have had available, others try to reduce the five ways to a single logical scheme, and others try to reduce all five ways to the four

causes (Wippel 2000, pp. 497–99, n. 161, n. 162, n. 163). Kenny (1969, p. 36) follows the last approach, claiming that "the five ways have in common a formal structure which in turn is applied to the four causes, and the different types of causality provide different contents for this formal structure" (De Haan 2013, pp. 147–52). In my view, Kenny is correct in claiming that the first two ways apply the notion of efficient causality, and the fifth way is associated with final causality. On the other hand, Kenny assumes that the third way is concerned with material causality and the fourth way argues from formal causality; this does not convince me. As analyzed, in the third way, Aquinas argues by inferring God, a necessary existent in itself, from necessary existents through another, which in turn cause possible existents. Neither possible existents nor necessary existents, whether they derive their necessity from another or not, can be equated with material cause. As for the fourth way, I argue that Aquinas does not argue from Aristotelian formal cause but from Platonic exemplar cause. Contrary to Wippel (2000, pp. 473–74), who identifies formal causality with exemplar causality, I suggest that the two appear similar but are essentially different. According to the Aristotelian four causes, material and formal causes, i.e., matter and form, are combined to form a sensible compound. From the Platonic perspective, the sensible compound is identified as a copy and comes into being through participation in the model, which refers to the exemplar cause. Many Thomists, including Kenny, have tried to show that Aquinas has an Aristotelian origin in constructing his five-way proof; nevertheless, Aquinas's sources of thought may be more than Aristotelian, as evidenced, for example, by his reworking of Avicenna's metaphysical proof in the third way and his adherence to Platonism in the fourth way. Therefore, Aquinas's five-way proof cannot be simply summarized by Aristotle's four causes.

Regarding the unified interpretation of Aquinas's five ways, I agree with Gilson (1957, pp. 76–77), who summarizes three features that the five ways have in common. First, Gilson notes that each of the five ways begins with the observation of sensible things because, according to Aquinas, an existence can be inferred only by proceeding from some other existence. As the five ways show, Aquinas deduces God's existence from some other existence, inferring God as the First Mover from the things He moves, God as the First Efficient Cause from the things He causes, God as the necessary existent in itself from the necessary existents that derive their necessity from another, God as the being in the highest degree from the things that exist in a higher or lower degree than others, and God as the intelligent being from the things that He directs toward the ideal. Gilson then mentions the other two features that the five ways have in common, which can be summarized as causality and the prohibition of infinite regression.[51] Given the prohibition of infinite regression, Aquinas deduces God as cause from the corresponding effect, so that God is described as the cause of all motion, the efficient cause of all things, the necessary existent in itself, the Supreme Being, and the ruler of all things. Aquinas thus follows the Aristotelian pattern of inferring cause from effect. Just as Aristotle deduces the First Mover from the moved heavenly bodies, Aquinas deduces God as Creator from the creatures (Wippel 2000, p. 497).

In addition, I offer an explanation of why Aquinas returns to the Aristotelian cosmological proof against the background of Avicenna's harsh criticism from three perspectives, one epistemological, one metaphysical, and one theological. First, Aquinas is able to construct the cosmological proof that proceeds from effect to cause because he follows the Aristotelian epistemological principle, according to which our cognitive activity proceeds from what is prior to us to what is prior by nature. By "what is prior to us", Aquinas does not mean what is prior to the human intellect (as Avicenna claims) but rather what is prior to sensation (as Aristotle asserts). What is prior to sensation is the sensible thing from which Aquinas begins his argument. Thus, in each of the five ways, Aquinas proceeds from the sensible, which is prior to us, to the intelligible, which is prior by nature, that is, from effect to cause (*ST* I, q. 2, a. 2). Second, Aquinas returns to the cosmological proof that Avicenna discarded because causality, that is, the causal relationship in the creationist sense between the creatures and the Creator, plays a crucial role in his metaphysical considerations.[52]

Aquinas replaces Avicenna's distinction between the possible existent and the necessary existent with the distinction of cause and effect in the creationist sense, i.e., the Creator and the creatures, because in Aquinas's view, Avicenna's distinction is merely conceptual, whereas the Creator and the creatures establish a real distinction. Although Avicenna's syllogistic way of deducing God as a special existent from a universal existent is necessary, in Aquinas's view, what is demonstrated by the syllogism is not a truly existing God but merely the concept of God. The same criticism applies to Anselm's ontological proof (*SCG* I, c. 11). To show that God actually exists, this leaves only cosmological proof, which follows the Aristotelian pattern of inferring cause from effect. Third, Aquinas constructs his cosmological proof by inheriting the Aristotelian tradition that lays the foundation for natural theology. Accordingly, humankind has access to God through our natural reason. Aquinas's theological thought includes not only natural theology but also negative and revelatory theology (sacra doctrina). In accordance with Aquinas's cosmological proof, natural theology holds that humankind can ascend to God through our natural reason, while negative and revelatory theology holds that it is possible for God to descend from His divine nature to humankind only through His own freedom and will. Although they cut off the path of human reason ascending from humankind to God, negative and revelatory theology preserve the path of grace descending from God to humankind, leaving room for mysticism in philosophy, salvation by grace in theology, and religious experience for believers.

In summary, Aquinas believes that God's existence cannot be inferred from any idea or concept but only from some other existence; therefore, he criticizes Anselm's ontological proof by the idea of God and abandons Avicenna's metaphysical proof by the concept of existence while restoring Aristotle's cosmological proof by causality. Just as Aristotle deduces God as the First Mover from the moved heavenly bodies, Aquinas constructs his five-way proof to deduce God as the Creator from the creatures, with the Creator appearing as the First Mover, the necessary existent in itself, or the efficient, final, or existential cause of all things. Aquinas returns to the Aristotelian cosmological proof, which proceeds from effect to cause, not only because he restores the Aristotelian epistemological principle, which proceeds from the sensible to the intelligible, but more fundamentally because causality, that is, the causal relationship in the creationist sense between the creatures and the Creator, plays a key role in his metaphysical considerations. More importantly, in constructing the five-way proof, Aquinas follows the Aristotelian tradition that lays the foundation for natural theology. Thus, on the basis of natural theology, human beings have the capacity to know God through natural reason.

## 5. Conclusions

I examine the three proofs of God's existence proposed by Aristotle, Avicenna, and Thomas Aquinas. Since philosophers and theologians dedicated to proving God's existence do not question God's existence, proving God's existence is primarily not an ontological issue but an epistemological issue that discusses how we as humans are able to know God. Thus, the proof of God's existence is first and foremost an epistemological question closely related to epistemological principles. According to the Aristotelian epistemological principle that proceeds from the sensible to the intelligible, Aristotle constructs a cosmological proof by inferring God as the First Mover from the moved heavenly bodies, and Aquinas constructs a five-way proof by inferring God as the Creator from the creatures. While Aristotle and Aquinas follow the same pattern of inferring cause from effect, Avicenna deduces God as a special existent from a universal existent in accordance with the epistemological principle of deduction from the universal to the particular. Aristotle, Avicenna, and Aquinas construct different proofs of God's existence not only because they follow different epistemological principles but, more fundamentally, because they construct metaphysics in different ways and emphasize different issues within their metaphysical frameworks. Aristotle constructs a cosmological proof because causality, that is, the causal relationship in the sense of moving and being moved between God and the heavenly bodies, plays a

crucial role in the metaphysical framework of ousiology and aitiology. Aquinas constructs a five-way proof, similar to Aristotle's cosmological proof, because causality, that is, the causal relationship in the creationist sense between God as Creator and the creatures, plays a key role in his metaphysical considerations. While Aristotle and Aquinas focus on causality (albeit in different senses), Avicenna focuses on modality. Avicenna constructs a metaphysical proof because modality, especially the mode of necessity, occupies a central place in his metaphysical construction (Bertolacci 2008; Adamson 2013; De Haan 2020), and more importantly, he uses syllogism to reconstruct metaphysics as a universal science, i.e., ontology. In addition, Aquinas restores the cosmological proof that Avicenna discards with his theological intention so that one can rationally address humankind's relationship with God. Consistent with the cosmological proof, natural theology holds that humankind has the capacity to know God through our natural reason. In conclusion, Aristotle's cosmological proof by causality, Avicenna's metaphysical proof by modality, and Aquinas's five-way proof by causality differ from each other because of their different epistemological principles, different metaphysical constructions, and particular theological intentions.

Finally, regarding the question of what proof of God's existence actually proves, I would offer the following explanation. Philosophers and theologians who are dedicated to proving God's existence do not question God's existence; they disagree on the way in which God exists. Therefore, what proof of God's existence does is to specify God's nature or essence because God's existence and essence are unified. Different proofs characterize God's essence differently. Aristotle uses a cosmological proof to establish God as the First Mover in terms of causality. Avicenna uses a metaphysical proof to establish God as the necessary existent in itself in terms of modality. In his five-way proof, Aquinas defines the nature of God not only in terms of modality as the necessary existent in itself (as Avicenna does) but also primarily in terms of causality (as Aristotle does), identifying God as the First Mover, the primary efficient, final, and existential cause. Thus, Gilson is correct in stating that Aquinas, proceeding from existence to essence, uses "proofs for the existence of God to form a notion of His essence" (Gilson 1957, p. 57; 2002, p. 51) and thus properly lists the proofs for God's existence under the general heading "On the Essentiality of the Divine Substance" (Gilson 1957, p. 52; 2002, p. 46). In my view, Gilson's statement applies not only to Aquinas's five-way proof but also to Avicenna's metaphysical proof and Aristotle's cosmological proof. It is through these proofs of God's existence that we finally arrive at the essence of God. In turn, the different essences of God that one arrives at through different approaches do not in any way undermine the existential uniqueness and simplicity of God.

**Funding:** This research was funded by the National Social Science Foundation of China, grant number 21BZX088.

**Institutional Review Board Statement:** Not applicable.

**Informed Consent Statement:** Not applicable.

**Data Availability Statement:** No new data were created or analyzed in this study. Data sharing is not applicable to this article.

**Conflicts of Interest:** The author declares no conflict of interest.

## Notes

1.    Aristotle, *Metaph*. Γ2, 1003a33-1003b19. I subscribe to the "focus-meaning" interpretation proposed by some interpreters; see Aubenque (1978, 1986) and Owen (1986).
2.    In many places (*Metaph*. B1, 995b13-15, 995b31-36; B2, 997a34-b3; Z2, 1028b27-32; M1, 1076a10-13), Aristotle asks whether there is a kind of substance other than sensible substance, i.e., intelligible substance. He rejects two possible answers by criticizing Plato's ideas and classifying mathematical entities as quantities. The only intelligible substance that Aristotle truly recognizes is God. I will explain this point in due course.
3.    Aristotle, *Metaph*. E1, 1026a27-29; K7, 1064b9-11; *PA* A1, 641a34-36.
4.    Aristotle, *Metaph*. E1, 1026a29-31; Z11, 1037a13-17; K7, 1064b11-14; M1, 1076a8-15; *Phys*. B2, 194b9-15; *PA* A1, 641a36-b4.
5.    Aristotle, *Metaph*. Λ1, 1069a36-b2; Λ10, 1075b130-14; K2, 1060a27-31.

6   In *Metaph.* Λ1, 1069a30-36, Aristotle speaks of the three kinds of substances (οὐσίαι δὲ τρεῖς, 1069a30), natural things, heavenly bodies, and God. Aristotle is precise and correct in stating that natural things such as (human beings), animals, and plants are sensible and perishable substances (1069a30-32). However, the (four) elements appear in place of the heavenly bodies, representing sensible and imperishable substances (1069a32-33), and the ideas and mathematical entities appear in place of God, representing intelligible and immovable substances (1069a33-36). According to Aristotle's doctrine of categories, mathematical entities are not substances, but belong to the category of quantity. Aristotle sharply criticizes Plato's ideas and does not recognize them as substances. Although they are intelligible and immovable, Plato's ideas and mathematical entities are not substances in the Aristotelian sense, so they should not appear here. Although he recognizes the four elements as substances in *On Generation and Corruption*, Aristotle does not discuss them in *Metaphysics* Λ, so they should not appear here. In this case, one might ask why Aristotle mentions the elements in Λ1, which he does not discuss here, and speaks of Plato's ideas and mathematical entities, which he does not consider substances. I suppose that Aristotle introduces these into his discussion of substances because he uses the doxographical method. In order to express his own doctrine, Aristotle presents the doctrine proposed by pre-Aristotelian philosophers and argues with them. To present the doctrine of the four causes, in *Metaph.* A, Aristotle goes through all the pre-Socratics, Socrates and Plato, who discuss the related issues. Similarly, in *Metaph.* Λ, Aristotle refers to the pre-Aristotelian doctrines of substances, although he disagrees with some of them, in order to present his own doctrine of substances. After introducing the three kinds of substances in Λ1, Aristotle discusses them one by one. He presents natural things as sensible and perishable substances in Λ2-5, heavenly bodies as sensible and imperishable substances in Λ6-8, and God as intelligent and immovable substance in Λ6-9. In particular, in Λ6-8 Aristotle presents God as the First Unmoved Mover by relating Him to the moved heavenly bodies, and in Λ9 he presents God as a purely intelligent entity. Finally, Aristotle summarizes the relationships among these three kinds of substances in Λ10. Thus, in terms of the overall structure and content of Book Λ, Aristotle discusses the three kinds of substances: natural things, heavenly bodies, and God. For Aristotle's discussion of these three substances, see also Liu (2019, pp. 24–25).

7   Aristotle, *Metaph.* Γ2, 1003b17-19; Λ1, 1069a25-26. See also Liu (2019, pp. 25–26).

8   Scholars have vigorously debated whether Aristotle's metaphysics is metaphysica generalis (i.e., ontology), metaphysica specialis (i.e., theology), or both, and I will not enter into that discussion here. Instead, I offer an alternative interpretation by suggesting that Aristotle constructs metaphysics as ousiology and aitiology. For a detailed discussion of how Aristotle constructs ousiology and aitiology and how he deals with the relationship between them, see Liu (2019, pp.1–33).

9   See also Aristotle, *Phys.* A1, 184a18-21; A5, 188b30-33, 189a4-9; *APo.* A1, 71b33-72a5; *De An.* B2, 413a11-13; *Metaph.* Δ11, 1018b29-34; Z3, 1029b3-12; *NE* A2, 1095a30-b4. See also Liu (2019, p. 26, n. 34).

10   Scholars disagree on the number of heavenly bodies Aristotle mentions in *Metaph.* Λ8, 1073b17-1074a14. Ross (1924, p. 392) thinks there are fifty-five heavenly bodies, while Cohen and Reeve (2020, p. 22) think there are ninety-four. I will not enter into this discussion here, because what is important for the cosmological proof is that there are many heavenly bodies, not one, and that there are finitely many, not infinitely many.

11   See also Aristotle, *Phys.* Θ5, 256a13-14; H1, 242a49-50.

12   Aristotle believes that the heavenly bodies cannot interact with each other because, in my interpretation, Aristotle sees the heavenly bodies as material entities that cannot actively move themselves or anything else without an external efficient cause.

13   Aristotle, *Metaph.* Λ7, 1072a24-26; *Phys.* Θ5, 256a4-21; H1, 242a49-66.

14   Alexander, *Alexandri In Metaphysica Commentaria* 686. 2–16.

15   Enrico (2001, p. 202): "Furthermore, says Theophrastus, if heaven is living like other living beings, its movement could be explained by the action of its soul, and would not need any unmovable mover."

16   Aristotle, *Metaph.* Λ8, 1074a36-37; *Phys.* Θ5, 256a13-21; H1, 242a49-55.

17   For a detailed reconstruction of Aristotle's cosmological proof and a detailed discussion of two assumptions of this proof, see Liu (2019, pp. 275–82).

18   See also Aristotle, *Metaph.* Λ7, 1072b18-30; Λ9, 1074b38-1075a10.

19   God constantly realizes Himself throughout eternity by driving the heavenly bodies in an eternal circular motion. The circular motion of the heavenly bodies, especially the motion of the sun around the earth, affects the creation and destruction of all natural things in the sublunar world. According to Aristotle's "geocentric theory", human beings, plants, and animals are created when the sun reaches its perigee; they are destroyed when the sun reaches its apogee. For a detailed discussion of how the three kinds of substances, that is, God, heavenly bodies, and natural things, are related to each other, see Aristotle, *Metaph.* Λ10, 1075a11-25; Liu (2019, pp. 287–91).

20   According to the traditional interpretation, Avicenna provides a proof of God's existence in *Ilāhīyyāt* I.6-7. Some scholars question the traditional interpretation and suggest that Avicenna does not prove God's existence until *Ilāhīyyāt* VIII.1-3 (Davidson 1987; Bertolacci 2007; De Haan 2013, 2016).

21   Some scholars see Avicenna's proof as cosmological (Davidson 1987; Chignell and Pereboom 2020), while others see it as metaphysical (Marmura 1980; Lasica 2019).

22     I use the terms "metaphysical" and "ontological" as synonyms for the proof of God's existence. While some scholars (Mayer 2001; Lasica 2019) characterize Avicenna's proof as ontological, I prefer to present it as metaphysical, so that Avicenna's metaphysical proof is not confused with Anselm's ontological proof.

23     Avicenna, *Commentaire sur le livre Lambda de la Metaphysique d' Aristote (chapitres 6–10)*, (2014, pp. 48–49); Gutas (2014, p. 199).

24     Honnefelder (1987, p. 168): "Was Metaphysik von Gott erkennen kann, vermag sie nach Aristoteles nur im Ausgang von den Wirkungen zu erkennen. Ein solcher Leitfaden erlaubt aber nur eine Erkenntnis Gottes per accidens."

25     Aristotle, *APo.* A6, 74b5-12, 75a12-14, 75a28-31; A7, 75a38-b6; A8, 75b21-24; A9, 76a4-15; A10, 76b11-16.

26     Avicenna, *Ilāhīyyāt* I.5, p. 22, lines 19-22. I refer to Marmura's and Bertolacci's translation with slight modification.

27     Avicenna, *Ilāhīyyāt* I.5, p. 22, lines 19-22; p. 23, lines 15–17; see also Bertolacci (2008, p. 36, n. 18).

28     Bertolacci (2006, p. 155): "As the fact of occurring at the very beginning of the *Ilāhīyyāt* witnesses, the articulation of Ontology constitutes, according to Avicenna, the main axis of metaphysics."

29     In the proemium of his commentary on Aristotle's *Metaphysics*, Aquinas defines metaphysics from three perspectives and gives it three names. Metaphysics is called first philosophy, which examines the primary causes of things; metaphysics is also called divine science or theology, which examines the intelligent being, completely separated from matter, both in reality and in the mind; and metaphysics is finally called transphysics, which examines the universal principles, being and its essential properties, such as one–many and potency–actualization (Metaphysica, inquantum considerat ens et ea quae consequuntur ipsum. Haec enim transphysica inveniuntur in via resolutionis, sicut magis communia post minus communia. [...] Unde et illa scientia maxime est intellectualis, quae circa principia maxime universalia versatur. Quae quidem sunt ens, et ea quae consequuntur ens, ut unum et multa, potentia et actus). In my opinion, Aquinas inherits the ontological conception of metaphysics from Avicenna, while he inherits the theological and aitiological conceptions of metaphysics from Aristotle.

30     Aristotle, *APo.* A4, 73b16-21; A10, 76b3-16; Avicenna, *Ilāhīyyāt* I.3; Bertolacci (2006, pp. 213–30; 2007, pp. 65–73).

31     Bertolacci (2008, pp. 48–49) does not call Avicenna's division diairesis, and I interpret his description and summary of Avicenna's division as diairesis.

32     According to the recent research, there is ample evidence that Avicenna not only recognizes cross-division but also applies it on a wide scale. Lammer has astutely noted that in the *Physics of the Healing* (al-Samāʾal-ṭabīʿī, 2009, p. 39), Avicenna classifies power (*quwwa*, *potentia*, δύναμις) into four types (Lammer 2018, pp. 290–99, 306). I believe that Avicenna establishes the fourfold classification by means of cross-division. Avicenna constructs the cross-division by cross-combining the two pairs of attributes, i.e., single function–multiple function and without volition–with volition, with each other. The cross-combination of the two pairs of attributes results in the four pairs of combinations that characterize four kinds of things endowed with a certain power: the natural thing is single-function and acts without volition, the celestial soul is single-function and acts with volition, the vegetative soul is multifunction and acts without volition, and the animal soul is multifunction and acts with volition. I would also note that Lammer has also mentioned another cross-division made by Themistius, Philoponus, and al-Fārābī, such that the two pairs of attributes, i.e., relative–absolute and hypothesis–postulate, are cross-combined. In this way, the fourfold classification is established: relative hypotheses, relative postulates, absolute hypotheses, and absolute postulates (Lammer 2018, pp. 88–91). Moreover, I observe that in *Ilāhīyyāt* I.2, Avicenna uses cross-division to divide mathematics into four subdisciplines and the subject of mathematics into four kinds. The subject of mathematics, quantity, is divided into two parallel pairs, continuous–discrete and abstraction from matter–existence in matter, and the two pairs are cross-combined, thus forming a cross-division that establishes four pairs of combinations that characterize four kinds of quantity. Through this cross-division, quantity, the subject of mathematics, is divided into four kinds, and accordingly, mathematics is divided into four subdisciplines as follows. Geometry studies the quantity that is continuous and abstract from matter (i.e., lines, surfaces, and bodies); astronomy studies the quantity that is continuous and exists in matter (i.e., heavenly bodies); arithmetic studies the quantity that is discrete and abstract from matter (i.e., numbers); and music studies the quantity that is discrete and exists in matter (i.e., notes). Based on what has been said, it is reasonable to assume that in *Ilāhīyyāt* I.6, Avicenna uses cross-division to make a fourfold classification of existent. Notably, Avicenna did not invent cross-division. Plato and Aristotle used cross-division extensively; in his commentary on Aristotle's *Categories*, Porphyry notes Aristotle's use of cross-division and calls it chiasmus (χιαστή). See Porphyry (1887) *Porphyrii Isagoge et In Aristotelis Categorias Commentarium*, 78.34-79.11; Liu (2019, pp. 15–18, n. 16, n. 17, n. 18). For Plato's and Aristotle's use of chiasmus and a detailed discussion of the difference between chiasmus and diairesis, see Liu (2021).

33     Some scholars claim that Avicenna inaugurated the second beginning of metaphysics by reconstructing metaphysics, although they disagree on how Avicenna made a new beginning for metaphysics. See Verbeke (1983, p. 10, 23), Bertolacci (2007, p. 73) and Aertsen (2012, p. 75).

34     Bertolacci (2007, p. 64): "Avicenna defends (...) that metaphysics can have a theological goal precisely because it has an ontological starting point".

35     Kenny, Shields, and Pasnau suggest that Aquinas uses movetur in both intransitive and passive senses; MacDonald, Wippel, and Pawl read movetur only in the passive sense. See Kenny (1969, pp. 8–9), MacDonald (1991, pp. 121–24), Wippel (2000, pp. 414–15, 444), Pawl (2012, pp. 116–17, 127, n. 18) and Shields and Pasnau (2016, pp. 107–14).

36     In Greek, however, κινεῖσθαι can also be used in the sense of a medium. Notably, Aristotle claims not that something moved must be moved by something else (*ab alio*) but rather that something moved must be moved by something (ὑπό τινος). Although

Aristotle's aim is to infer an external mover from the moved heavenly bodies, his formulation and the possible use of κινεῖσθαι in the sense of medium indicate that Aristotle does not completely exclude the internal efficient cause and emphasize the external efficient cause, as Aquinas does. Thus, there is a slight difference between Aquinas's imitation and Aristotle's original formulation, although this difference does not affect the argument.

37    Thanks to one of the anonymous reviewers who mentions Aristotle's definition of motion. From this perspective, the matter is more clearly illuminated. Aristotle defines motion as the actualization of what exists potentially, insofar as it exists potentially (*Phys.* Γ1, 201a10-11). In other words, motion is defined as the process from potentiality to actuality. At the beginning, the thing exists in potentiality, and motion does not begin; at the end, the thing achieves its goal and exists in actuality, and motion ends. Movement is neither beginning nor end, neither potentiality nor actuality, but the process from beginning to end and from potentiality to actuality. From the perspective of movement, it is illuminating that potentiality and actuality cannot coexist simultaneously. According to Aristotle, however, not only can moving and being moved coexist, but they must operate simultaneously for movement to occur (*Phys.* Γ3, 202a21-b5). For example, the moving hand and the moved stick operate simultaneously to lift the stick. The same reasoning applies to the movement of animals: the moving soul and the moved body work simultaneously for a human being to act. When there is movement, the moving and the moved work together at the same time. Therefore, I think the two sets, potentiality–actuality and moving–being moved, cannot be confused with each other.

38    In my opinion, the argument proposed by Aquinas in the *SCG* I, c. 13 bears much resemblance to the argument established by Proclus in the *Elements of Theology* (abbreviated *ET*). In the seventeenth proposition, Proclus argues, "But if the mover is one part and the moved another, the whole will not in itself be self-moved, since it will be composed of parts that are not self-moved: it will have the appearance of being self-moved, but in essence it will not be so" (*ET*, Prop. 17. lines 5-8). I have slightly modified Dodd's translation.

39    Aristotle, *Metaph.* Λ3, 1070a7-8; *Phys.* B1, 192b8-15, 27–30.

40    Accordingly, in explaining the four causes, Aquinas regards the material and formal causes as internal, while he treats the efficient and final causes as external; see *De principii naturae* c. 3 (1999, p. 60): Causas autem accipit tam pro extrinsecis quam pro intrinsecis: Materia et forma dicuntur intrinsecae rei, eo quod sunt partes constituentes rem; efficiens et finalis dicuntur extrinsecae, quia sunt extra rem.

41    Gilson (1957, pp. 66–68; 2002, p. 64) emphasizes that in the second way, Aquinas places the hierarchical, vertical structure among the efficient causes in order to avoid infinite regression. Gilson (2002, p. 74) further asserts that "all the proofs presume that the causes and effects appearing in them are arranged hierarchically". Pasnau takes up this idea by claiming that in each of the five ways, Aquinas replaces the infinite horizontal series of causes going back in time with a vertical series, thus ending the infinite regression. See Shields and Pasnau (2016, p. 112) and Pasnau (2022, p. 5).

42    Pawl (2012, p. 122) also questions the validity of this deduction.

43    In explaining that the truest being is the Supreme Being, scholars carefully distinguish between logical truth and ontological truth (what Shields and Pasnau call the "ontic conception/sense of truth"), correctly noting that Aquinas uses "truth" not in the logical sense (e.g., a true proposition) but in the ontological sense (i.e., truth refers to reality). See Wippel (2000, p. 471), Pawl (2012, p. 124) and Shields and Pasnau (2016, pp. 116–17).

44    Pawl, Shields, and Pasnau interpret Aquinas's dictum literally, suggesting that Aquinas is appealing to Aristotle, as he claims in the fourth way; see Pawl (2012, p. 124) and Shields and Pasnau (2016, pp. 114–15).

45    Plato, *Soph.* 254b-260a.

46    Plato, *Resp.* 595c-597e; *Tim.* 27c-31b, 39e-40d.

47    See (O'Rourke 1971) "Via causalitatis; via negationis; via eminentiae (Weg der Ursächlichkeit; Weg der Negation; Weg des Übermaßes)", in *Historisches Wörterbuch der Philosophie*, pp. 1034–38.

48    See also Proclus, *ET*, Prop. 8. lines 3-4: "If all beings desire the Good, it is evident that the First Good transcends [all] beings" (εἰ γὰρ πάντα τὰ ὄντα τοῦ ἀγαθοῦ ἐφίεται, δῆλον ὅτι τὸ πρώτως ἀγαθὸν ἐπέκεινά ἐστι τῶν ὄντων). I have slightly modified Dodds's translation.

49    To track down Aquinas's resources for constructing the fourth way, I make a strict distinction between Platonism and Neoplatonism, the distinction between via eminentiae and via negationis, and carefully emphasize that here, in the fourth way, Aquinas takes the via eminentiae by following Platonism. Influenced by Pseudo-Dionysius, Aquinas generally uses not only the positive way but also the negative way to characterize God. Thanks to the anonymous reviewer for reminding me of this point.

50    MacDonald (1991) focuses on the first way, Shields and Pasnau (2016) on the first and fourth ways. According to Wippel and Pawl, many scholars focus on the third way; see Wippel (2000, pp. 462–63, n. 52, 464, n. 57, 465, n. 60, n. 61, 466, n. 63, 466–67, n. 64) and Pawl (2012, p. 130, n. 49).

51    In the sixth and final edition and translation of *Le Thomisme*, Gilson (2002, p. 74) summarizes the last two features common to the five ways as one characteristic by stating that "a second characteristic is that all the proofs presume that the causes and effects appearing in them are arranged hierarchically". In other words, the second common feature is both causality and the hierarchical order between cause and effect. Gilson (2002, pp. 74–75) emphasizes the hierarchical order between cause and effect because only the hierarchy can end the infinite regression.

[52] Aquinas not only emphasizes causality, i.e., the causal relationship between the Creator and the creatures, in the five-way proof, he also emphasizes it from other perspectives. Since the Creator absolutely transcends and completely dominates the creatures, the causal relationship between them is asymmetrical, irreversible, and creationist. To express the causal relationship between the Creator and creatures in this sense, Aquinas uses the analogy of attribution in such a way that the same predicate "good" is said of God and creatures, such as "God is good" and "creatures are good". The predicate "good" is not used univocally because God is essentially different from creatures; nor is it used purely equivocally because, by creating them, God is closely related to creatures. Since God is both essentially different from and causally related to creatures, the relationship between God and creatures can be adequately expressed only with the analogy of attribution, and "good" is thus used analogically. Within the analogy of attribution, Aquinas distinguishes between the analogy of many-to-one and the analogy of one-to-another, with the latter characterizing the relationship of creatures to God; e.g., the goodness of creatures is caused by and oriented toward the absolute goodness of God. Aquinas also characterizes this kind of relationship with participation, according to which creatures are good because they participate in the absolute goodness of God. Creatures can participate in any property of God because God has made it all possible. In order to accurately and properly characterize the causal relationship in the creationist sense between the Creator and creatures, Aquinas introduces participation and the analogy of attribution. Thus, as we have seen, many of Aquinas's important metaphysical considerations, such as participation, analogy, and the five-way proof of God's existence, revolve around causality in the creationist sense between the Creator and the creatures. On Aquinas's analogy, see *De potentia* q. 7, a. 7; *SCG* I, c. 29-36.; *ST* I, q. 13, a. 5-6. For an interpretation of Aquinas's analogy, see Montagnes ([1963] 2004), McInerny (1996), Pannenberg (2007), Wippel (2000), Aertsen (2012), Spencer (2015), and Hochschild (2019).

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
