# Peer review of "On Proofs for the Existence of God: Aristotle, Avicenna, and Thomas Aquinas"

_religions, doi:10.3390/rel15020235_

Round 1
Reviewer 1 Report
Comments and Suggestions for Authors
This paper discusses the various God proofs of Aristotle, Avicenna and Thomas Aquinas. In the interpretation given, the emphasis is on the epistemological side of the God proofs. The author shows how Thomas Aquinas beyond Avicenna reverts to the cosmological proof formulated by Aristotle. In doing so, Thomas Aquinas distances himself from Avicenna's conceptualist approach.
This paper is relatively voluminous but nevertheless has a very nice style in which no topic is discussed for too long and the reader can keep a certain steady pace. This manifests itself, for example, where perhaps an explanation would be required, where the author gives as his conviction that Thomas Aquinas commits an error (12, 476). Creditable is the excellent way in which the author summarizes the various God proofs. The way he chooses from the extensive scholarly literature on the subject is also very good.
In a general sense, I would say that in my opinion the author's original contribution to the existing discussion is somewhat understated. What exactly is the point the author wants to make? We have difficulty in understanding this from the very neutral title of the paper, nor from the research question or the conclusion. Clear communication is of the essence here.
The last paragraph of the conclusion should be revisited by the author. Does this conclusion really fit the whole of the paper? To answer the question of what a God proof actually proves, the author would have to study more of its authors than he actually did in this paper. To explain this with Thomas Aquinas: Thomas makes a clear distinction between existence (an sit) and being (quid sit) of God, and limits the meaning of his God proofs to the former. This is evident both from the introductory text to the God Proofs in the Summa Theologiae and from the sequel. Surely, then, it is not very plausible that both the conclusion and the summary (p. 18) ignore this distinction. The statement "Thus, on the basis of natural theology, human beings have the capacity to attain communion with God through natural reason" seems to me to be simply incorrect as far as Thomas Aquinas is concerned. Leaving aside the fact that Thomas has no "natural theology," and the God proofs themselves are included in the larger picture of his version of "sacra doctrina," real communion with God is a matter of grace for Thomas and nothing else.
Another passage that may need to be revisited by the author is the first paragraph of page 16 on the Platonic rather than Neo-Platonic nature of Thomas' fourth God Proof. The author contrasts the "absolutely negative way" of describing God (Neo-Platonism) with the "absolutely positive way" of Platonism preferred by Thomas, according to him. In the first place, to speak of 'absolutely' seems to me misplaced, given the background of the triplex via of Pseudo-Dionysius. But more importantly, why would Thomas make this choice, when you consider that a) he clearly writes in the prologues of quaestiones 2 and 3 that people (without help) cannot know how God is and what God is, but only how and what He is not, and b) both in 12.2 and 12 and 13.9 he makes this 'negative' approach his own, and c) puts himself behind the Neo-Platonic triplex via (e.g. in 13.3 ad 2). Again, then, the author's failure to engage with the context in which the God proofs are placed by Thomas seems to bar a correct interpretation. In any case, the speculative interpretation given by the author here does not strike me as very plausible.
I would like to recommend revising the abstract. Now it seems that the theme of philosophy-theology is a very important theme in this paper, but it is not. Moreover, the sentence "Moreover, proof of God's existence is of theological significance" remains rather up in the air. At the same time, the abstract also reveals that this paper lacks a clear focus when it comes to the unique contribution the author wants to make to the discussion on this topic.
Editorial:
- P. 12, 463: omit the comma after ‘For Aquinas’
- P. 16, note 46: ‘Weg der Negaton‘ should be ‘Weg der Negation‘
- P. 7, 281-283: is this a quotation? This should be made clearer.
Reviewer 2 Report
Comments and Suggestions for Authors
Reviewer 3 Report
Comments and Suggestions for Authors
The paper's comparison of Aristotle's, Avicenna's and Aquinas' proofs for the existence of God is quite good. The author achieves most of what he or she sets out to do. The author does show how these proofs differ from each other because of their different epistemological principles and metaphysical constructions. The author falls short of showing how the various proofs clarify the essence of God, largely because it would take a longer paper to achieve that task.
The accounts of Aristotle's and Aquinas' proofs are accurate, with a few exceptions. Since I am less familiar with Avicenna, I will defer to other reviewers the accuracy of the author's claims about him.
The account of Aristotle's cosmological proof is very good, but there are two points with which I take issue. In footnote 6, the author claims that in Metaphysics XII Aristotle presents natural things as sensible and perishable substances in XII.2-5, sensible and imperishable substances in XII.6-8, and God in XII.9. But this is inaccurate. God is also discussed in XII.6-8 (as some of the author's other footnotes acknowledge, e.g., footnotes 13 and 16). Since the footnote 6 remark is inaccurate, revise it.
My second concern about the author's claims about Aristotle comes up in the author's account of Aquinas' First Way. Now, the account of Aquinas' Five Ways is generally good, but the author relies on Shields and Pasnau to claim that, according to Aristotle, "all natural [emphasis mine] things (such as plants, animals, and humans). . . can move themselves" (p. 12, lines 479-480). At best, the author should say "all living things..." Rocks do not move themselves. But even with that correction, I am not sure that Aquinas is reworking Aristotle's philosophy by distinguishing between self-motion per se and self motion per accidens. It seems to me that Aristotle is making a similar point about animal motion in On the Soul III.10.433b10-30.
Furthermore, given the Aristotelian definition of motion employed in the First Way (the fulfillment of what exists in potentially, in so far as its exists potentially--Physics III.1.201a10-11), I am unconvinced that "Aquinas confuses two sets of concepts here by substituting the non-coexistence of potentiality and actuality (2.2.) for the non-coexistence of moving and being moved (2.3)" (p. 12, lines 476-478).
My last two objections are minor. First, the author contends that proofs for God's existence are an epistemological and not an ontological issue (pp. 18-19). This claim is unconvincing; they can be both. Second, the author claims that "natural theology holds that humankind has the capacity to attain communion with God through our natural reason" (p. 19, lines 790-791). What does the term "communion" mean? Knowledge of God as an object of theoria? Friendship? The author needs to clarify.
Comments on the Quality of English Language
There are two minor typos that need to be corrected. The heading for section 4 should read "Thomas Aquinas' [not Aquina's] Cosmological Proof for the Existence of God" and 'Shields' is misspelled in footnote 35.
I also find the numbering for the premises in the First Way (p. 13, lines 441-460) confusing. Why is there a (2), (2.1), and (2.2) and a (3.1.), (3.2), but no (3)?
